# Which Parameter Influences Local Disease-Free Survival after Radiation Therapy Due to Osteolytic Metastasis? A Retrospective Study with Pre- and Post-Radiation Therapy MRI including Diffusion-Weighted Images

**DOI:** 10.3390/jcm11010106

**Published:** 2021-12-25

**Authors:** Jiyeong Lee, Young Cheol Yoon, Ji Hyun Lee, Hyun Su Kim

**Affiliations:** Department of Radiology, Samsung Medical Center, Sungkyunkwan University School of Medicine, Seoul 06351, Korea; zzunge0125@naver.com (J.L.); carrot302@hotmail.com (J.H.L.); calmuri@naver.com (H.S.K.)

**Keywords:** radiation therapy, bone metastasis, diffusion-weighted imaging, apparent diffusion coefficient, histogram, local disease progression-free survival

## Abstract

Although radiation therapy (RT) plays an important role in the palliation of localized bone metastases, there is no consensus on a reliable method for evaluating treatment response. Therefore, we retrospectively evaluated the potential of magnetic resonance imaging (MRI) using apparent diffusion coefficient (ADC) maps and conventional images in whole-tumor volumetric analysis of texture features for assessing treatment response after RT. For this purpose, 28 patients who received RT for osteolytic bone metastasis and underwent both pre- and post-RT MRI were enrolled. Volumetric ADC histograms and conventional parameters were compared. Cox regression analyses were used to determine whether the change ratio in these parameters was associated with local disease progression-free survival (LDPFS). The ADC_maximum,_ ADC_mean,_ ADC_median,_ ADC_SD_, maximum diameter, and volume of the target lesions after RT significantly increased. Change ratios of ADC_mean_ < 1.41, tumor diameter ≥ 1.17, and tumor volume ≥ 1.55 were significant predictors of poor LDPFS. Whole-tumor volumetric ADC analysis might be utilized for monitoring patient response to RT and potentially useful in predicting clinical outcomes.

## 1. Introduction

Bone metastases, which occur in up to 70% of cancer patients [1], are a major cause of morbidity, including bone pain, impaired mobility, pathologic fractures, hypercalcemia, and spinal cord compression, all of which can severely impair quality of life [2]. Therapeutic goals in patients with bone metastases are to delay progression, alleviate symptoms, improve quality of life, and obtain any possible survival benefit [3]. The important role of radiation therapy (RT) in the palliation of localized bone metastases is well acknowledged, with its intent to reduce tumor growth and improve symptom control [4]. To determine the optimal management to minimize radiation dose and prevent recurrence, it is important to evaluate the response to treatment [5]. However, assessing the treatment response with conventional images is difficult because the healing process of bone metastases is slow to evolve and subtle [2,6]. To date, there has been no consensus on a reliable method for evaluating the treatment response, making therapeutic decisions difficult [7,8].

In addition to conventional sequences, magnetic resonance imaging (MRI) can provide functional information on cellularity and molecular activity using diffusion-weighted imaging (DWI) [9]. Because malignant lesions differ in their cellularity and biological aggressiveness, DWI is increasingly being used in the context of bone marrow evaluation of metastatic disease [10,11], and an apparent diffusion coefficient (ADC) map derived from DWI enables us to quantitatively assess the treatment response [12]. Several studies have reported the potential of MRI with DWI for assessing treatment response after RT for bone metastases [5,13,14]. However, the previous reports either evaluated a single section of the lesion or used only mean ADC values, which could result in intratumoral sampling bias or could not reflect tumor heterogeneity. This drawback may be overcome by whole-tumor volumetric and texture analyses. Histogram texture analysis can supply a quantitative methodology using every voxel of the tumor [15,16].

Therefore, the primary objective of the present study was to evaluate the differences in parameters from anatomical images and ADC maps using whole-tumor volumetric analysis of texture features between pre- and post-RT MRI in patients with osteolytic metastases. Additionally, we explored whether the change ratios of MRI-derived parameters have a prognostic value for the prediction of local disease progression-free survival (LDPFS).

## 2. Materials and Methods

### 2.1. Patients

This retrospective study was approved by our institutional review board (SMC 2020-05-024). From an oncology database at our institution between August 2012 and May 2019, 273 patients, who were diagnosed with bone metastasis by histological or clinicoradiological confirmation, underwent RT with or without chemotherapy. The clinicoradiological diagnosis was made using two prerequisites: typical imaging features (such as a new osteolytic or contrast-enhancing lesion and an increase in the size of the lesion) and progression in size and number during the follow-up period before RT in patients with known primary malignant tumors. The inclusion criteria for patients were as follows: (1) patients who underwent MRI including DWI at baseline (within 1 month prior to starting treatment) and at about 6 months (150–180 days) after completion of RT (decided arbitrarily after considering other previous studies [17,18,19]) and (2) patients who had metastasis in the pelvic and appendicular bones, with the exclusion of spine MRI due to different protocols in our institute. The exclusion criteria were as follows: (1) osteoblastic or mixed bone metastases; (2) prior history of RT, chemotherapy before RT, or metallic instrumentation at the metastatic sites; (3) inadequate MRI follow-up; and (4) pathological conditions such as fracture or infection on MRI after RT. Figure 1 illustrates the patient selection process.

Finally, 28 patients were enrolled in the study. All clinical data, including age, sex, primary cancer, and RT dose, were retrospectively obtained from medical records.

### 2.2. MRI Protocols

All patients underwent MRI examination using 3.0-T MRI scanners (Ingenia; Philips Medical Systems, Best, The Netherlands) prior to initiating RT (pre-RT) and within 6 months (post-RT; 150–180 days) after RT. The MRI protocol included turbo spin-echo (TSE) axial T1-weighted (T1WI) and T2-weighted (T2WI) images, sagittal T2WI, and coronal T1WI images as conventional MRI sequences. For DWI, axial single-shot echo-planar imaging was acquired using sensitizing diffusion gradients in the x, y, and z directions and *b* values of 0, 400, and 1400 s/mm^2^, according to a previous study on the optimization of the *b* value for bone marrow imaging [20]. The DWI consisted of 20 transverse sections with a section thickness of 4 or 5 mm. The ADC maps were automatically generated from the DWI using commercial diffusion analysis software (Extended MRI workspace, version 2.6.3.1. Philips Healthcare). Contrast-enhanced axial and coronal T1WI were acquired after intravenous injection of contrast material (gadoterate meglumine; Dotarem^®^, Guerbet, Roissy, France; 0.1 mmol/kg body weight by power injector).

### 2.3. Image Analysis

All pre- and post-RT MRIs were independently analyzed by two board-certified radiologists (readers I and II, with 5 years and 1 year of experience in musculoskeletal MRI, respectively) using a software package (EXPRESS, Philips Korea, Seoul, Korea) for whole-tumor volume analysis of the ADC map, with the aid of a picture archiving and communication system (PACS; Centricity, GE Healthcare, Chicago, IL, USA) for anatomical reference, without any knowledge of the clinical information. They drew the volume of interest (VOI) on the ADC map with the aid of conventional image sets if the boundary of the target lesion was not clearly delineated. The maximum diameter, which was defined as the longest diameter among the standard axial, coronal, or sagittal planes, was measured using the PACS system. Whole-tumor volume and ADC-driven parameters (minimum, maximum, mean, median, standard deviation (SD), skewness, and kurtosis) were calculated from the VOI. If a patient had multiple bone metastases, the largest lesion was selected.

### 2.4. Treatment Response Evaluation

According to the MD Anderson (MDA) criteria, local tumor response was evaluated (complete response (CR), partial response (PR), progressive disease (PD), and stable disease (SD)) [21]. PD was defined as follows: (1) ≥25% increase in the sum of the perpendicular diameters of any measurable lesion on radiography, computed tomography (CT), or MRI or (2) ≥25% subjective increase in the size of unmeasurable (such as ill-defined) lytic lesions on radiography, CT, or MRI. By comparing images at the time point within 1 month before RT (baseline) and the time to progression during serial follow up, both readers categorized the patients into PD or non-PD (CR, PR, and SD) groups with a consensus, at which time same-modality images were used. LDPFS was defined as the time between baseline and follow-up images, which showed PD according to the MDA criteria.

### 2.5. Statistical Analysis

The ADC parameters derived from histogram analysis included minimum, maximum, mean, median, SD, skewness, and kurtosis (ADC_minimum_, ADC_maximum_, ADC_mean_, ADC_median_, ADC_SD_, ADC_skewness_, and ADC_kurtosis_, respectively). Changes in MRI-driven parameters were defined as the ratio of values after RT to values before RT, by dividing the value of post-RT MRI by that of pre-RT MRI (*_R_*ADC_minimum_, *_R_*ADC_maximum_, *_R_*ADC_mean_, *_R_*ADC_median_, *_R_*ADC_SD_, *_R_*ADC_skewness_, *_R_*ADC_kurtosis_, *_R_*Tumor diameter, and *_R_*Tumor volume). The Wilcoxon signed-rank test and paired t-test were used to compare MRI-driven parameters before and after RT. For dichotomization, the optimal cutoff values of parameters were determined at the point where the log-rank p value was at a minimum [22]. Kaplan–Meier curves were compared using the log-rank test. Cox regression analyses were used to determine whether changes in these parameters and clinical variables such as age, cancer type, RT dose, and metastatic site were associated with LDPFS. 

Interobserver agreement was evaluated using the intraclass correlation coefficient (ICC). The ICC values were determined to represent slight agreement (0.00–0.20), fair agreement (0.21–0.40), moderate agreement (0.41–0.60), substantial agreement (0.61–0.80), almost perfect agreement (0.81–0.99), and perfect agreement (1.00) [23]. Retrospective power analysis was performed by using the paired-t test. Statistical significance was set at *p* < 0.05. Statistical analyses were performed using SAS software (version 9.4; SAS Institute), IBM SPSS Statistics (version 27.0; IBM Corp, Armonk, NY, USA), and MedCalc Statistical Software (version 19.4.0; MedCalc Software Ltd., Ostend, Belgium).

## 3. Results

A total of 28 patients (16 men and 12 women; mean age 60.5 years, range 44–80 years) were enrolled in this study (Table 1).

The interobserver agreement between the two readers for the measurement of MRI parameters was as follows: ADC_minimum_, ICC = 0.704 (95% confidence interval (CI) 0.527–0.820); ADC_maximum_, ICC = 0.931 (95% CI 0.883–0.960); ADC_mean_, ICC = 0.986 (95% CI 0.976–0.992); ADC_median_, ICC = 0.984 (95% CI 0.973–0.991); ADC_SD_, ICC = 0.967 (95% CI 0.944–0.980); ADC_skewness_, ICC = 0.835 (95% CI 0.735–0.900); ADC_kurtosis_, ICC = 0.829 (95% CI 0.719–0.897); maximum diameter, ICC = 0.986 (95% CI 0.976–0.992); tumor volume ICC = 0.990 (95% CI 0.984–0.994). The measurement of all MRI parameters except ADC_min_ showed almost perfect interobserver agreement; thus, the measurements of reader I were used. 

### 3.1. Comparison of MRI-Derived Parameters between Pre- and Post-RT

Among the ADC parameters, the ADC_maximum_, ADC_mean_, ADC_median_, and ADC_SD_ of target lesions after RT showed a significant increase (*p* < 0.001), with a change of +0.56 × 10^−3^ mm^2^/s ± 0.49, +0.38 × 10^−3^ mm^2^/s ± 0.36, +0.37 × 10^−3^ mm^2^/s ± 0.38, +0.12 × 10^−3^ mm^2^/s ± 0.17 (mean ± SD), respectively (Table 2). As conventional parameters, the maximum diameter and volume of target lesions also significantly increased after RT (+0.2 ± 0.7 cm, +6.1 ± 18.6 cm^3^, (mean ± SD), *p* values; 0.018, 0.001, respectively). Although not statistically significant, kurtosis values tended to decrease after treatment (−1.26 ± 3.51, (mean ± SD), *p* = 0.051). The change in skewness was not statistically significant. Power analysis revealed that power reached >0.999 and 0.819 for change of ADC_mean_ between pre- and post-RT at a significance level of 0.05 and 0.0001, respectively.

### 3.2. Associations between Range-Ratio of MRI Parameters and Local Disease Progression-Free Survival (LDPFS)

The median LDPFS was 20 months (range, 1–63 months). The cutoff values for *_R_*ADC_mean_, *_R_*ADC_SD_, *_R_*ADC_skewness_, *_R_*ADC_kurtosis_, *_R_*Tumor diameter, and *_R_*Tumor volume were determined to be 1.41, 1.03, 0.56, 0.73, 1.17, and 1.55, respectively. Patients with *_R_*ADC_mean_ < 1.41 (log-rank *p* = 0.0243), *_R_*ADC_SD_ < 1.03 (log-rank *p* = 0.0499), *_R_*Tumor diameter ≥ 1.17 (log-rank *p* = 0.0024), and *_R_*Tumor volume ≥ 1.55 (log-rank *p* = 0.0070) had shorter LDPFS than patients with *_R_*ADC_mean_ ≥ 1.41, *_R_*ADC_SD_ ≥ 1.03, *_R_*Tumor diameter < 1.17, and *_R_*Tumor volume < 1.55, respectively (Figure 2).

Table 3 presents the outcomes of the Cox regression analyses affecting LDPFS. Because of the too small ratio of events per variable in the study, multivariable analysis was not performed [24]. Univariable analysis demonstrated that *_R_*ADC_mean_ < 1.41 (hazard ratio (HR) = 3.817, *p* value = 0.036), *_R_*Tumor diameter ≥ 1.17 (HR = 5.802, *p* value = 0.007), and *_R_*Tumor volume ≥ 1.55 (HR = 5.155, *p* = 0.016) were significant prognostic factors for predicting poor LDPFS. Figure 3 and Figure 4 display representative examples of patients in the non-PD and PD groups.

## 4. Discussion

We investigated changes in the ADC parameters derived from whole-tumor volumes of bone metastases after RT and evaluated their association with LDPFS. Our results demonstrated that the ADC_maximum_, ADC_mean_, ADC_median_, and ADC_SD_ significantly increased 6 months after RT. Additionally, the ratios of change in ADC_mean_, tumor diameter, and tumor volume were significant prognostic factors predicting LDPFS.

ADC is inversely correlated with tissue cellularity [25]. Increased ADC values indicate an increase in extracellular water content and loss of cell membrane integrity, whereas decreased ADC values represent a decrease in extracellular water or increase in cell number or size [26]. Various studies have suggested that ADC values increase after treatment and have demonstrated the potential of ADC evaluation for monitoring response after chemotherapy or RT [5,13,14,17,18,19,27]. We carried out whole-tumor volumetric ADC histogram analysis. Previous studies demonstrated that if tumors respond successfully to treatment, due to post-treatment changes (such as tumor necrosis or a reduction in cell density), kurtosis values generally decrease, the standard deviation increases, and skewness often develops a negative value (tail to the left) [28,29,30]. Likewise, our study showed a significant increase in ADC_SD_ values, in addition to ADC_maximum_, ADC_mean_, and ADC_median_ after treatment and their potential use in treatment response assessment. Although not statistically significant, kurtosis values tended to decrease after RT, which was presumed to be related to the limitations of our study, which will be covered later. After receiving effective treatment, ADC values were distributed more heterogeneously, reflecting the necrotic or hemorrhagic regions within the tumor. Visually, the changes reflected in the ADC histogram turned into a wider spread (increased SD) and lower peak (decreased kurtosis). 

Although there have been various studies on the potential of changes in ADC values for monitoring treatment response, to the best of our knowledge, there have been no studies on the potential for predicting clinical outcomes in bone metastases. Several studies demonstrated that changes between pre- and post-treatment ADC parameters were correlated with treatment response and clinical outcome in malignant brain tumors [31,32] and pancreatic cancer [33]. Accordingly, it seems reasonable to assume that there may be associations between changes in ADC parameters after RT and local disease progression in bone metastases. Our results highlighted that a less than 41% increase in ADC_mean_ was a significant predictor for poor LDPFS. Therefore, an ADC_mean_ value lower than the cutoff value is considered to be associated with a poor prognosis. Due to lack of multivariable analysis, confounding variables were not controlled in this study, which is one of limitations in this study. However, the fact that these potential confounders including age, cancer type, RT dose, and metastatic site were not significantly associated with LDPFS on univariable analysis might mitigate their confounding effects, although they are not removed.

The present study had several limitations. First, the data were analyzed retrospectively; as a result, several clinical data were intentionally out of the scope of this study, such as various primary tumor types and treatment protocols (extent of radiation dose, adjuvant chemotherapy, or hormone therapy). In addition, we could not monitor the follow-up period after RT. Second, only a relatively small number of patients could be included because most patients in this disease setting had already received either chemotherapy or RT prior to the initial MRI. This small sample size also prohibited the evaluation of multivariable analysis, as described above. Third, using 0 s/mm^2^ as the first *b* value instead of 50 s/mm^2^ might increase the contribution of blood perfusion to ADC measurements [34]. Fourth, a validation study could not be performed. 

## 5. Conclusions

In summary, ADC parameters (ADC_maximum_, ADC_mean_, ADC_median_, ADC_SD_) significantly increased 6 months after RT. *_R_*ADC_mean_ < 1.41, *_R_*Tumor diameter ≥ 1.17, and *_R_*Tumor volume ≥ 1.55 in the 6 months post-RT MRI compared to the pre-RT MRI were significant prognostic factors for predicting poor LDPFS. Our results suggest that whole-tumor volumetric ADC analysis might be utilized for monitoring patient response to RT and potentially useful in predicting clinical outcomes.

## Figures and Tables

**Figure 1 jcm-11-00106-f001:**
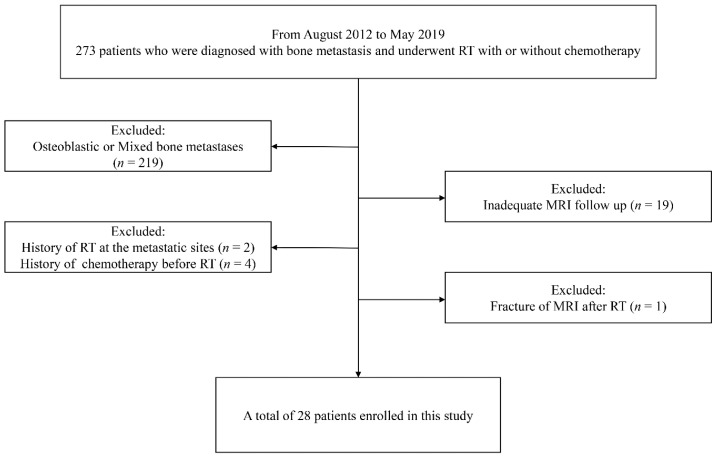
Patient selection flow chart. RT, radiation therapy.

**Figure 2 jcm-11-00106-f002:**
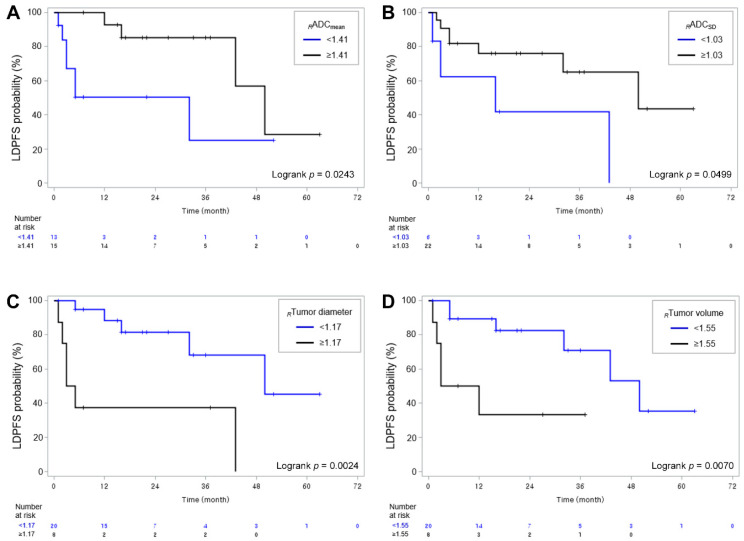
Kaplan–Meier curves showing LDPFS of patients according to (**A**) *_R_*ADC_mean_, (**B**) *_R_*ADC_SD_, (**C**) *_R_*Tumor diameter, and (**D**) *_R_*Tumor volume with their cutoff values. LDPFS, local disease progression-free survival; ADC, apparent diffusion coefficient; SD, standard deviation.

**Figure 3 jcm-11-00106-f003:**
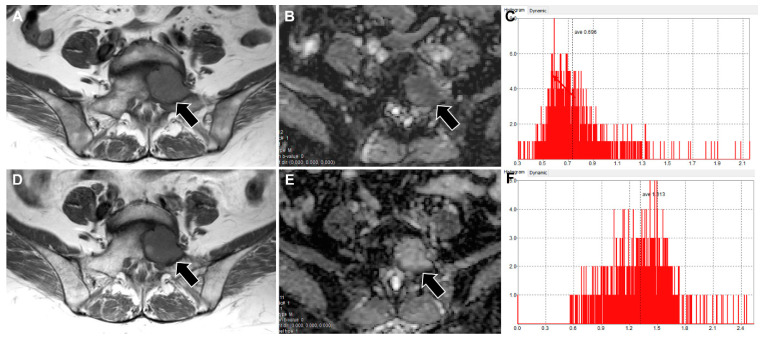
Images of a 72-year-old woman with metastasis to the S1 vertebra from a thyroid cancer. (**A**,**D**) T1WI, (**B**,**E**) ADC maps, and (**C**,**E**) histograms showing the pixel distribution of ADC values in selected VOI from the pre-RT (**A**–**C**) and post-RT (**D**–**F**) MRI are shown. Compared with pre-RT MRI, the mean and SD of ADC values of post-RT MRI were increased from 0.70 to 1.31 (×10^–3^ mm^2^/s) and 0.21 to 0.29, respectively, with _R_ADC_mean_ and _R_ADC_SD_ calculated to be 1.87 and 1.38, respectively. The sums of the greatest diameters and volumes were measured to be 6.6 cm and 10.9 cm^3^, respectively, in pre-RT, and 5.9 cm and 10.8 cm^3^, respectively, in post-RT MRI, which were not significantly changed. Local disease progression did not occur during the follow-up period of 36 months. T1WI, T1-weighted image; ADC, apparent diffusion coefficient; VOI, volume of interest; RT, radiation therapy; SD, standard deviation.

**Figure 4 jcm-11-00106-f004:**
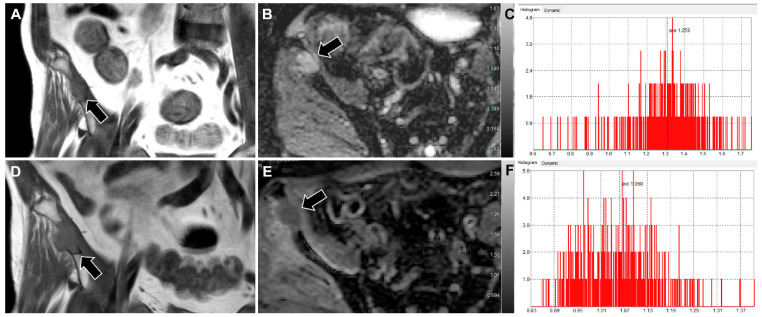
Images of a 63-year-old man with metastasis to the right iliac bone from a hepatocellular carcinoma. (**A**,**D**) Coronal T1WI, (**B**,**E**) ADC maps, and (**C**,**E**) histograms showing the pixel distribution of ADC values in selected VOI from the pre-RT (**A**–**C**) and post-RT (**D**–**F**) MRI are shown. Compared with pre-RT MRI, the mean and SD of ADC values of post-RT MRI were decreased from 1.26 to 1.06 (×10^–3^ mm^2^/s) and 0.18 to 0.10, respectively, with _R_ADC_mean_ and _R_ADC_SD_ calculated to be 0.84 and 0.56, respectively. The sum of greatest diameters increased from 5.9 to 8.7 cm, representing PD. T1WI, T1-weighted image; ADC, apparent diffusion coefficient; VOI, volume of interest; RT, radiation therapy; SD, standard deviation; PD, progressive disease.

**Table 1 jcm-11-00106-t001:** Patients’ characteristics.

Patient	Sex	Age (yr)	Target Lesion Location	Primary Cancer Type	Radiation Dose (cGy)	Follow-Up Time (Months)	Sum of Greatest Diameters * (cm)		Last FU Image	Disease Status
							Pre RT	Post RT		
1	F	72	Sacrum	Thyroid cancer	5000	36	6.6	5.9	CT	Non-PD
2	M	80	Pelvic bone	Lung cancer	4500	7	10.9	13.2	CT	Non-PD
3	M	72	Humerus	Hepatocellular carcinoma	4300	15	7.0	6.0	CT	Non-PD
4	F	65	Hand	Renal cell carcinoma	4500	22	3.7	4.2	XR	Non-PD
5	F	45	Pelvic bone	Breast cancer	3750	63	3.0	2.1	CT	Non-PD
6	M	66	Foot	Renal cell carcinoma	4000	5	5.2	5.7	CT	Non-PD
7	M	70	Femur	Thyroid cancer	4500	52	3.0	2.9	MR	Non-PD
8	M	66	Clavicle	Hepatocellular carcinoma	5000	33	6.0	6.0	CT	Non-PD
9	F	63	Pelvic bone	Lung cancer	2500	7	3.8	4.5	CT	Non-PD
10	M	51	Foot	Hepatocellular carcinoma	3500	37	11.1	6.6	MR	Non-PD
11	M	59	Sacrum	Lung cancer	2000	5	10.7	12.4	MR	Non-PD
12	M	52	Pelvic bone	Cholangiocarcinoma	2400	17	14.6	15.2	XR	Non-PD
13	F	44	Femur	Rhabdomyosarcoma	3000	27	5.7	2.7	CT	Non-PD
14	M	56	Pelvic bone	Hepatocellular carcinoma	3600	22	3.9	4.2	CT	Non-PD
15	F	65	Sacrum	Thymic cancer	2800	1	6.6	8.8	MR	Non-PD
16	F	51	Humerus	Lung cancer	2500	21	5.4	1.4	MR	Non-PD
17	F	62	Pelvic bone	Thyroid cancer	3000	16	5.9	6.0	MR	Non-PD
18	F	56	Femur	Breast cancer	3000	16	6.9	9.0	MR	PD
19	F	72	Femur	Thyroid cancer	5000	32	3.1	3.9	MR	PD
20	M	48	Pelvic bone	Hepatocellular carcinoma	5000	12	3.5	6.3	MR	PD
21	F	60	Pelvic bone	Thyroid cancer	4500	50	3.1	3.9	MR	PD
22	M	67	Pelvic bone	Nasopharyngeal cancer	2000	3	3.2	5.0	MR	PD
23	M	55	Pelvic bone	Lung cancer	3000	3	15.8	19.8	XR	PD
24	F	57	Pelvic bone	Breast cancer	5400	43	4.4	6.2	CT	PD
25	M	63	Pelvic bone	Hepatocellular carcinoma	3000	5	5.9	8.7	MR	PD
26	M	63	Pelvic bone	Hepatocellular carcinoma	5000	1	3.8	5.5	MR	PD
27	M	57	Pelvic bone	Hepatocellular carcinoma	5000	2	4.1	5.4	MR	PD
28	M	56	Femur	Hepatocellular carcinoma	4000	5	9.7	14.6	MR	PD

* The sum of the perpendicular and bidimensional measurements of the greatest diameters of each individual lesion. CT, computed tomography; F, female; M, male; MRI, magnetic resonance imaging; PD, progressive disease; RT, radiation therapy; XR, radiography; FU, follow up.

**Table 2 jcm-11-00106-t002:** MRI parameters before and after RT.

	Baseline MRI	Post-RT MRI	*p* Value
ADC parameters			
Minimum (×10^−3^ mm^2^/s) *	0.37 (0.00–0.99)	0.53 (0.00–1.61)	0.120
Maximum (×10^−3^ mm^2^/s) ^#^	1.68 (0.85–2.55)	2.18 (1.08–3.18)	<0.001
Mean (×10^−3^ mm^2^/s) ^#^	0.95 (0.62–1.58)	1.33 (0.61–2.05)	<0.001
Median (×10^−3^ mm^2^/s) ^#^	0.94 (0.61–1.64)	1.31 (0.58–2.07)	<0.001
Standard deviation *	0.19 (0.07–0.48)	0.31 (0.07–0.89)	<0.001
Skewness ^#^	0.29 (−0.86–3.28)	0.10 (−0.98–1.74)	0.329
Kurtosis *	4.57 (2.49–19.19)	3.3 (1.42–7.67)	0.051
Conventional parameters			
Maximum diameter (cm) *	4.1 (1.5–10.5)	4.4 (1.6–13.0)	0.018
Volume * (cm^3^)	17.4 (0.8–111.0)	23 (0.9–203.9)	0.001

MRI, magnetic resonance imaging; ADC, apparent diffusion coefficient. Numbers are means with ranges in parentheses. * Wilcoxon signed-rank test. ^#^ Paired t test.

**Table 3 jcm-11-00106-t003:** Cox regression analysis for relation between MRI-derived parameters and local disease progression.

	HR	95% CI	*p* Value
Age	0.997	0.934–1.063	0.918
Cancer type	0.815	0.075–8.856	0.367
RT dose	1.123	0.573–2.198	0.736
Metastatic site *	2.100	0.550–8.020	0.278
*_R_*ADC_mean_ < 1.41	3.817	1.088–13.514	0.036
*_R_*ADC_SD_ < 1.03	3.311	0.924–11.905	0.066
*_R_*ADC_skewness_ < 0.56	6.211	0.732–52.632	0.094
*_R_*ADC_kurtosis_ < 0.73	0.361	0.077– 1.684	0.195
*_R_*Tumor diameter ≥ 1.17	5.802	1.604–20.989	0.007
*_R_*Tumor volume ≥ 1.55	5.155	1.357–19.582	0.016

CI, confidence interval; HR, hazard ratio; RT, radiation treatment; ADC, apparent diffusion coefficient; SD, standard deviation. * Metastatic site: pelvic bone (non-pelvic bone as a reference).

## Data Availability

Not applicable.

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
