# Peer review of "Which Parameter Influences Local Disease-Free Survival after Radiation Therapy Due to Osteolytic Metastasis? A Retrospective Study with Pre- and Post-Radiation Therapy MRI including Diffusion-Weighted Images"

_jcm, 2021, doi:10.3390/jcm11010106_

Round 1

Reviewer 1 Report

I thank the authors and the editor for this very interesting manuscript. 
However, I think the English should be read by a native-speaking.
The authors note the major limitations of this preliminary work. In particular, the very small size of the population and the heterogeneity of the patients and treatments. The greatest limitation is the absence of a validation cohort. 
Moreover, I think that it is difficult with such a small number of patients to carry out statistical tests of survival. In my opinion, the authors should limit themselves to univariate non-parametric statistical tests.

Reviewer 2 Report

The authors submitted the manuscript titled with” Which Parameter Influences the Local Disease Free Survival After Radiation Therapy Due to Osteolytic Metastasis? A Retrospective Study with Pre- and Post-radiation Therapy MRI Including Diffusion-weighted Images. Although the manuscript is well-written and well discussed, the novelty is limited due to a small cohort population containing with the heterogeneity of tumor types and osteolytic sites, as shown in the limitation.

There are several reasons to be clarified.

  1. To serve as a prognostic predictor, the change of interested parameters 6M after treatment seems to be outdated.
  2. Does the osteolytic site matter? Pelvic bone is almost half of the studied sites. Can these be categorized into pelvic or non-pelvic bone?
  3. Owing to small number, the result may be immature.
